# Enterotoxigenic *Escherichia coli* heat-labile toxin drives enteropathic changes in small intestinal epithelia

Enterotoxigenic *E. coli* (ETEC) produce heat-labile (LT) and/or heat-stable (ST) enterotoxins, and commonly cause diarrhea in resource-poor regions. ETEC have been linked repeatedly to sequelae in children including enteropathy, malnutrition, and growth impairment. Although cellular actions of ETEC enterotoxins leading to diarrhea are well-established, their contributions to sequelae remain unclear. LT increases cellular cAMP to activate protein kinase A (PKA) that phosphorylates ion channels driving intestinal export of salt and water resulting in diarrhea. As PKA also modulates transcription of many genes, we interrogated transcriptional profiles of LT-treated intestinal epithelia. Here we show that LT significantly alters intestinal epithelial gene expression directing biogenesis of the brush border, the major site for nutrient absorption, suppresses transcription factors HNF4 and SMAD4 critical to enterocyte differentiation, and profoundly disrupts microvillus architecture and essential nutrient transport. In addition, ETEC-challenged neonatal mice exhibit substantial brush border derangement that is prevented by maternal vaccination with LT. Finally, mice repeatedly challenged with toxigenic ETEC exhibit impaired growth recapitulating the multiplicative impact of recurring ETEC infections in children. These findings highlight impacts of ETEC enterotoxins beyond acute diarrheal illness and may inform approaches to prevent major sequelae of these common infections including malnutrition that impact millions of children.

Infectious diarrhea remains a leading cause of death and morbidity among young children in low-middle-income countries where access to clean water and sanitation remains in short supply[1]. Enterotoxigenic *E. coli* (ETEC), initially discovered as a cause of severe, cholera-like illness[2], are one of the most common pathogens associated with moderate-severe diarrhea among children under the age of 5 years[3,4], and are perennially the most common cause of diarrhea in travelers[5] to endemic regions where these organisms are thought to account for hundreds of millions of cases of diarrheal illness each year[6].

Importantly, ETEC infections have been linked to non-diarrheal sequelae, including "environmental enteric dysfunction (EED)," a condition characterized by impaired nutrient absorption, impaired growth[7,8], and malnutrition[9,10], adding significantly to the morbidity as well as deaths from diarrhea and other infections[11]. The risk of stunting multiplies with each episode of diarrheal illness in children under the age of two years[12], a period during which children residing in impoverished areas commonly sustain multiple ETEC infections[8]. However, the molecular pathogenesis underlying the intestinal changes associated with EED, and the contribution of individual pathogens, including ETEC, remain poorly understood.

Similarly, toxin-producing *E. coli* have also been repeatedly identified in patients with tropical sprue[13–15], a condition classically described in adults residing for extended periods of time in areas where ETEC diarrheal disease is common. Like EED, tropical sprue is

✉ e-mail: jfleckenstein@wustl.edu

associated with changes to the small intestinal villous architecture, including ultrastructural alteration of the epithelial brush border formed by the microvilli[16,17], nutrient malabsorption, and wasting.

The basic molecular mechanisms underpinning acute watery diarrhea caused by ETEC are well-established[18]. ETEC produces heat-labile (LT) and/or heat-stable (ST) enterotoxins that activate the production of cAMP and cGMP second messengers, respectively, leading to activation of cellular kinases that in turn modulate the activity of sodium and chloride channels in the apical membrane of intestinal epithelial cells to promote net efflux of salt and water into the intestinal lumen resulting in watery diarrhea.

LT and cholera toxin (CT) share ~85% amino acid identity, and both toxins exert their major effects on the cell through the ADP-ribosylation of the alpha subunit of Gs (Gsα), a stimulatory intracellular guanine nucleotide-binding protein. Inhibition of Gsα GTPase activity leads to constitutive activation of adenylate cyclase and increased production of intracellular cAMP[19].

In its central role as a second messenger, cAMP governs a diverse array of cellular processes[20] and modulates the transcription of multiple genes through a number of cAMP-responsive transcriptional activators and repressors[21]. cAMP activates protein kinase A (PKA), a heterotetramer, by liberating its two regulatory subunits from the catalytic subunits, which are then free to phosphorylate a wide variety of cytoplasmic and nuclear protein substrates[22]. PKA largely regulates transcription by phosphorylation of transcription factors, including the cyclic AMP response element binding protein (CREB) and the cAMP-response element modulator (CREM), which bind cAMP-response elements (CRE) in the promoter regions of target genes[21–23].

Notably, cholera toxin (CT), LT, and dibutyryl-cyclic AMP all induce hypersecretion and impact the architecture of gastrointestinal epithelia in rodent small intestine[24], while small intestinal biopsies of patients with acute cholera exhibit marked changes in the ultrastructure of the intestinal brush border, the major absorptive surface in the small intestine[25,26], including shortening and disruption of the microvilli. Consistent with these observations, studies of young children less than two years of age in Bangladesh have specifically asso-ciated LT-producing ETEC with undernutrition[27], suggesting that heat-labile toxin may exert effects on intestinal mucosa that extend beyond acute diarrheal illness.

Here we demonstrate that in addition to the canonical effects of LT on the cellular export of salt and water into the intestinal lumen, this toxin impacts multiple genes involved in the formation of microvilli, resulting in marked alteration of the architecture of the intestinal brush border, the major site of nutrient absorption in the small intestine. These effects are compounded by the alteration of solute transporters within the brush border epithelia, potentially disrupting the absorption of multiple essential nutrients.

## Results

### Heat-labile toxin markedly alters the transcriptomes of intestinal epithelial cells

Commensurate with the importance of cAMP as a second messenger, we found that compared to either untreated controls or cells treated with a catalytically inactive (E211K) mutant of LT, wild-type LT holotoxin substantially modulated transcription of many genes in intestinal epithelial cells. In RNA-seq studies of polarized Caco-2 intestinal epithelial cells, we found that 3832 genes were significantly ($p \leq 10^{-5}$) upregulated and 3687 downregulated in response to LT, while the inactive toxin failed to induce significant changes in the transcriptome (Fig. 1a). However, Caco-2 cells are derived from distant metastases of a colon cancer tumor in which transcriptomes would likely be altered relative to untransformed intestinal epithelia[28,29], and cAMP signaling is known to be aberrant in some transformed cells[23]. Therefore, to examine a more physiologically relevant target, we next examined the impact of LT on differentiated small intestinal enteroids. Here, we found that far fewer genes were differentially expressed ($\leq 10^{-5}$), with 746 significantly upregulated, and 561 downregulated in response to intoxication with LT (Fig. 1b). Notably, however, we found substantial statistically significant overlap in genes significantly modulated in Caco-2 cells and enteroids (Supplementary Table 3 and Supplementary Dataset 1) with the transcription of hundreds of genes significantly up- or down-regulated in both groups. Gene ontology enrichment analysis,

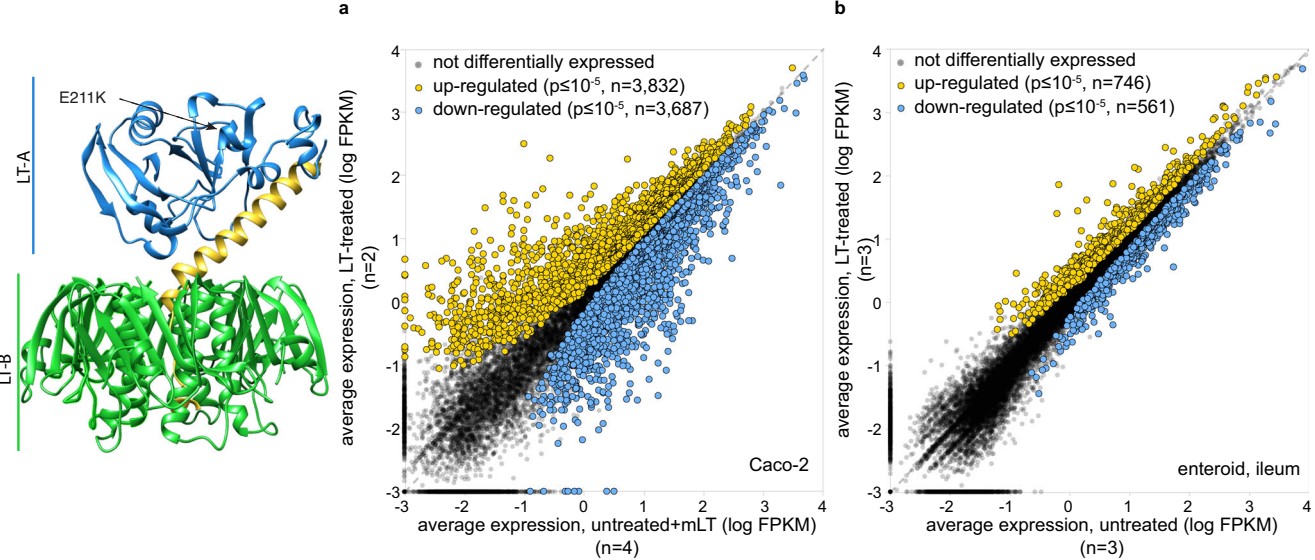

**Fig. 1 | Heat-labile toxin modulates expression of multiple genes in intestinal epithelia.** Model at left depicts the *E. coli* heat-labile toxin[92] based on PDB structure 1LTS with the A1 subunit in blue, the A2 region in yellow, and pentameric B subunit in green. The E211K mutation of mLT is in the active site of the A1 subunit. **a** Scatterplot of RNAseq data right depicts differential expression profiles of Caco-2 cells following exposure to a heat-labile toxin ($n = 2$) relative to untreated cells ($n = 2$) and cells treated with the biologically inactive mLT ($n = 2$). (Because expression profiles of untreated and mLT-treated cells were virtually identical, their combined expression profiles totaling $n = 4$ replicates are compared here to LT-treated cells). **b** RNA-seq data from polarized small intestinal ileal enteroids treated with LT ($n = 3$) compared to control untreated ($n = 3$) cells. Differentially expressed genes were identified by DESeq2[93].

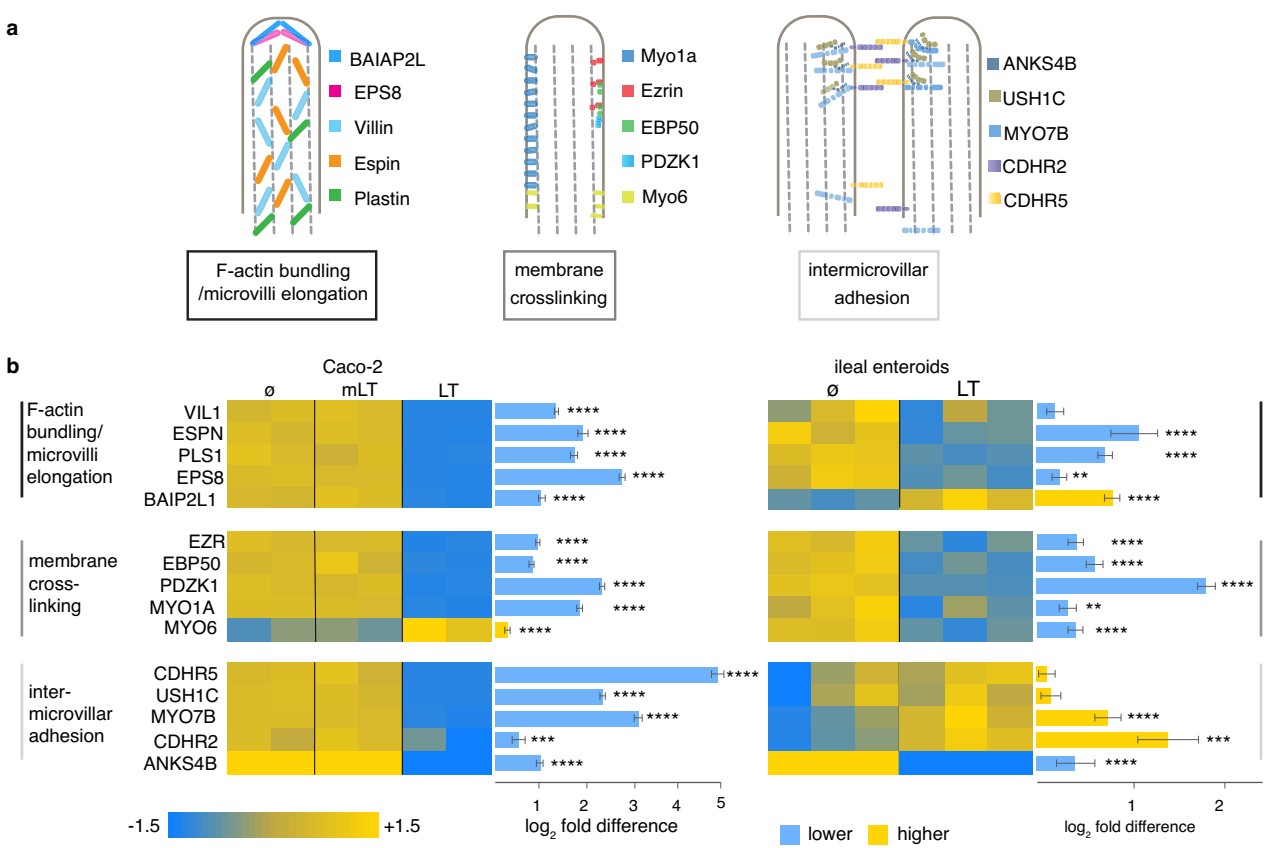

**Fig. 2 | Heat-labile toxin modulates multiple genes involved in microvillus assembly. a** Diagram at the top (adapted from ref. 30) depicts molecules involved in key elements of microvillus development. **b** Heatmaps of RNA-seq data obtained following treatment of Caco-2 intestinal cells (left) with mLT ($n = 2$ biologically independent samples) or LT ($n = 2$) relative to untreated cells ($n = 2$); and ileal enteroids (right) treated with LT ($n = 3$) relative to control untreated cells ($n = 3$). Comparisons were made with DESeq2[93]. Bars indicate absolute $\log_2$ fold change values + SE. *$p \leq 0.05$, **$p \leq 0.01$, ***$p \leq 0.001$, ****$p \leq 10^{-4}$, and *****$p \leq 10^{-5}$.

as well as ontology-independent investigation of genes (CompBio, Supplementary Fig. 1) modulated by the toxin, highlighted multiple cellular components associated with both the development and function of the absorptive surface of the small intestine (Supplementary Dataset 2).

## Heat-labile toxin impairs the development of small intestinal microvilli

Small intestinal enterocytes are each covered with hundreds of microvilli, complex structures comprised of a central core of actin filaments within protrusions of the plasma membrane. Collectively, the luminal surface of the intestine formed by these microvilli, known as the brush border, represents the major absorptive surface of the gastrointestinal tract. Three major classes of proteins are required for the biogenesis of microvilli[30] (Fig. 2a). These include (1) proteins such as villin, epsin, plastin, and EPS8 that bundle parallel clusters of actin filaments; BAIP2L1 (IRTKS) responsible for recruiting the EPS bundling protein to the tips of microvilli[31]; (2) ezrin, myo1a[32], myo6 that link the actin cytoskeleton with the plasma membrane; and (3) protocadherin molecules CDHR2 and CDHR5 engaged in extracellular heterotypic complexes between the tips of the microvilli[30] that are stabilized by a tripartite complex of MYO7B, ANKS4B[33], and USH1C[33]. Interrogation of transcriptional profiles indicated that the transcription of each of these classes of genes was significantly altered following exposure to wild-type heat-labile toxin (Fig. 2b). Similarly, RT-PCR confirmed decreased expression of multiple genes involved in microvilli biogenesis, including VIL1 encoding villin (Fig. 3a), and we were able to demonstrate that production of villin was depressed in polarized

small intestinal enteroids (Fig. 3b and Supplementary Fig. 2). In addition, TEM images of polarized small intestinal enteroids exposed to heat-labile toxin demonstrated significantly shortened and disorganized microvillus structures on the apical surface of enterocytes (Fig. 3c).

## Heat-labile toxin modulates the transcription of multiple brush border nutrient transport genes

The human solute carrier (SLC) gene superfamily is comprised of more than 50 gene families thought to encode more than 300 functional transporters[34]. Many of the SLC proteins are enriched in the small intestinal brush border, where they transport critical nutrients, including amino acids, oligopeptides, sugars, and vitamins. We found that transcription of many SLC genes was altered in Caco-2 cells as well as small intestinal enteroids (Fig. 4a). These included transporters for Zinc, known both to be deficient in children with enteropathy[35], and a micronutrient critical for intestinal homeostasis. Likewise, transcription of SLC19A3 encoding the principal SLC responsible for uptake of the water-soluble B vitamin thiamine (vitamin B1)[36] by differentiated intestinal epithelial cells lining the surface of intestinal villi of the proximal small intestine[37,38] (Fig. 4b) was repressed as was the production of the corresponding protein (supplementary figure 3a). Moreover, we found that LT treatment of human small intestinal organoids also interfered with the transcription of the cis-regulatory element specificity protein 1 (SP1) previously shown to govern the transcription of SLC19A3[39–41] (Fig. 4c and Supplementary Fig. 3b). Finally, we found that thiamine transport was significantly depressed following exposure to LT (Fig. 4d) providing additional evidence that ETEC can impair transport of critical nutrients.

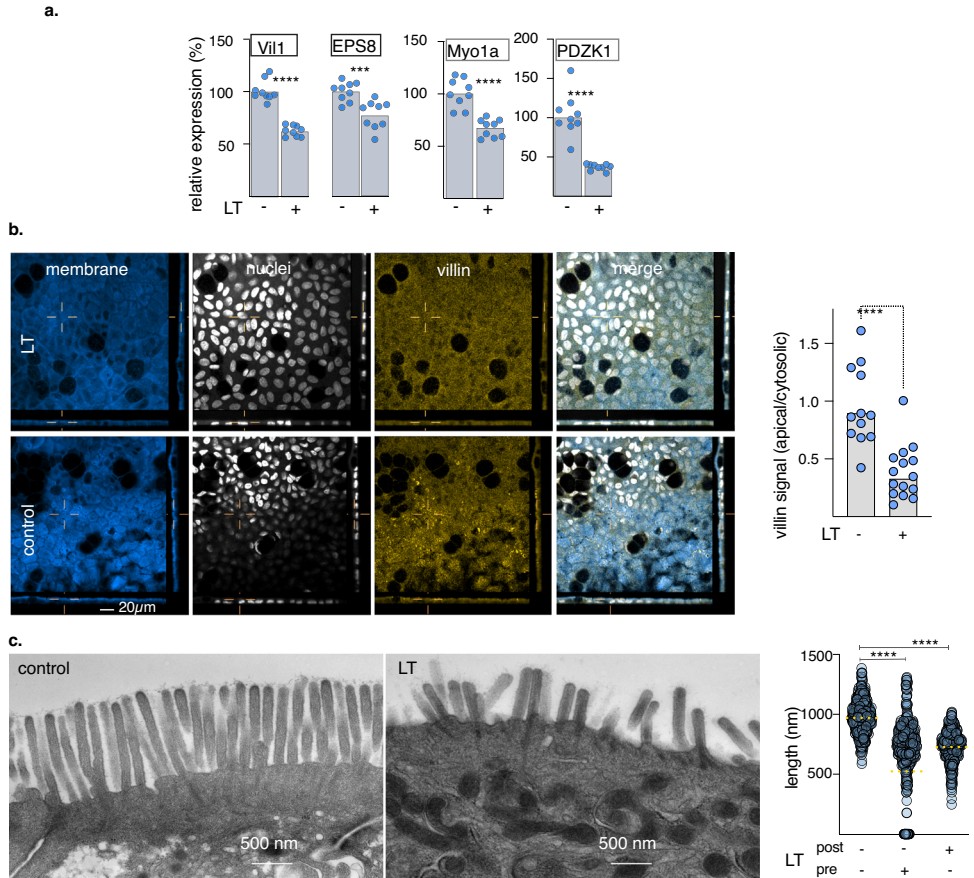

**Fig. 3 | Heat-labile toxin impairs the effective formation of small intestinal microvilli. a** Quantitative RT-PCR data from *n* = 9 biologically independent samples for selected microvillus genes comparing untreated (−) and LT-treated (+) ileal epithelial cells (enteroid line 235D). **b** Villin production is suppressed in small intestinal enteroids following treatment (t + 18 h) with LT (100 μg/ml). Shown are representative confocal images obtained showing membrane (CellMask, blue), nuclei (white), villin (gold), and merged image. The graph at right shows apical villin geometric mean fluorescence intensity data relative to the corresponding

cytoplasmic signal. Each symbol (*n* = 12) represents a unique region of interest. For **a**, **b** ****$p < 0.0001$, ***$p < 0.001$ by Mann–Whitney two-tailed testing. **c** TEM images of small intestinal microvilli following treatment LT (right) compared to control untreated cells (left). The graph at right shows the length of microvilli when enteroids (*n* = 2 biologically independent samples) are treated before (pre) and after (post) differentiation on polarized ileal cells ****<0.0001 by ANOVA (Kruskal–Wallis, nonparametric testing).

## LT-producing ETEC disrupts the absorptive architecture of the small intestine

To further study the potential impact of ETEC toxins on intestinal architecture, we performed challenge studies in infant mice. Compared to sham-challenged (PBS) controls, or mice challenged with a toxin-deficient strain of ETEC, we again noted down-regulation of genes involved in F-actin bundling, membrane cross-linking, and intermicrovillus adhesion complex formation, all required for intestinal microvilli (Fig. 5a) biogenesis. Likewise, on examination of small intestinal villi we found that the production of villin in enterocyte brush borders was substantially decreased relative to sham-challenged controls (Fig. 5b, c). In mice challenged with a wild-type ETEC isolate that makes ST and LT, but not an LT/ST-toxin-negative mutant (jf4763, Supplementary Table 1), we observed significant alteration in the architecture of the intestinal brush border with significant shortening and disorganization of the microvilli (Fig. 5d, e and Supplementary Fig. 4a–d) reminiscent of the earlier ultrastructural studies of patients with tropical sprue[17] and *V. cholerae* infections[25]. Similarly, we found that in mice challenged with a strain containing an isogenic mutation in *eltA* (jf571, Supplementary Table 1) encoding the LT A subunit, which still makes heat-stable toxins, the microvillus architecture was preserved (Fig. 5f, g), suggesting that LT is the principal toxin underlying the enteropathic changes to the enterocyte surfaces.

Despite the dramatic toxin-dependent changes to the absorptive surface of the intestine, the early growth kinetics of suckling mice challenged a single time with either wild-type bacteria or a heat-labile toxin deletion mutant were surprisingly similar (Supplementary Fig. 5A) and paralleled those of sham-challenged controls. However, enteropathy in young children is thought to reflect damage elicited by repeated infections. Children in endemic regions typically suffer *multiple* ETEC infections before their second birthday, and the risk of enteropathic sequelae increases multiplicatively per episode[8,12]. Therefore, to assess the contribution of repeated ETEC infections to growth impairment, we compared the growth kinetics of suckling mice challenged a single time to those repeatedly infected with wild-type toxigenic ETEC. These studies demonstrated a clear impact of repeated infection on growth (Supplementary Fig. 5B). Finally, we found that the growth kinetics of mice repeatedly challenged with wild-type ETEC H10407 was significantly retarded relative to those challenged with the isogenic LT-mutant jf876 (Supplementary Fig. 5C). Therefore, repeated infections in this model appear to recapitulate impacts observed following repeated ETEC infection in children, and our data suggest that these features are at least in part driven by LT.

## Maternal vaccination with LT prevents brush border disruption

To address whether enteropathic changes to the small intestine can be prevented by vaccination, and to further define the role of LT, we

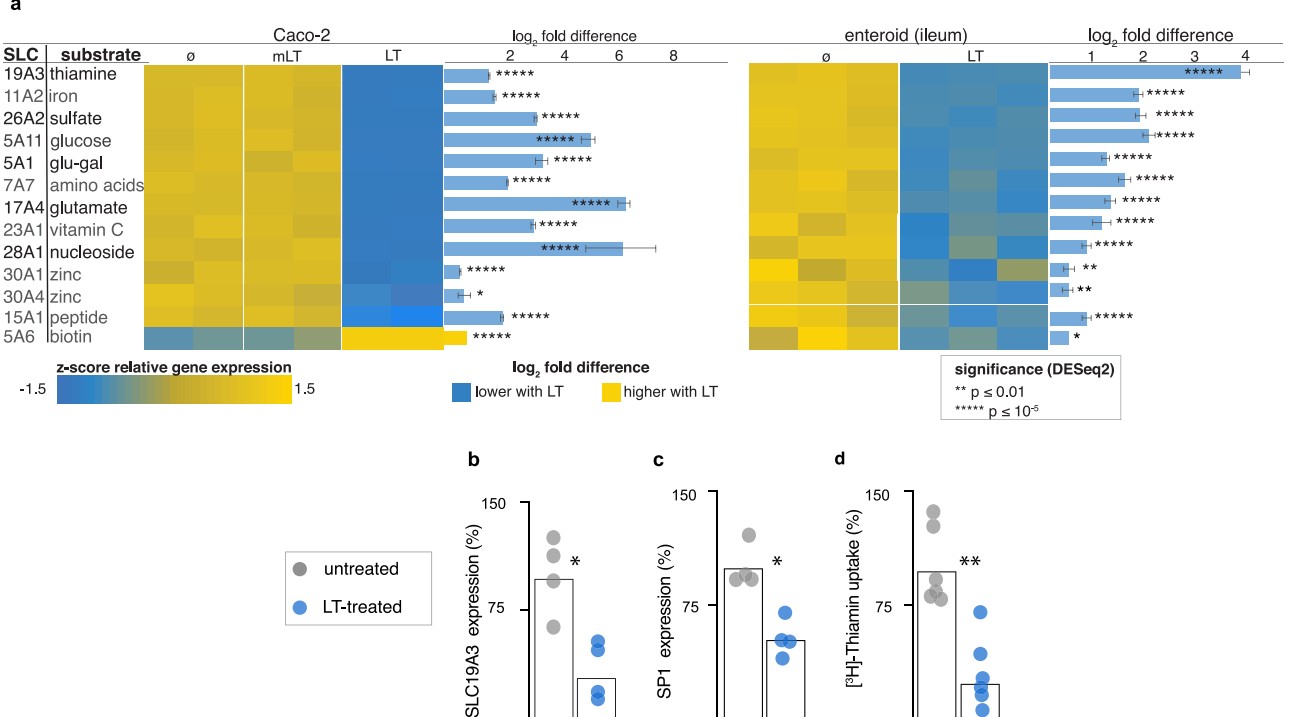

**Fig. 4 | Heat-labile toxin alters the transcription of multiple brush border SLC genes. a** Heatmap indicating key SLC genes modulated by heat-labile toxin (LT) compared to enzymatically inactive E112K LT mutant (mLT), or untreated (ø) Caco-2 cells (left) and human small intestinal (ileal) enteroids (Hu235D, right). Bars indicate absolute $\log_2$ fold change values + SE. Comparisons were made with DESeq2[93]. *$p \leq 0.05$, **$p \leq 0.01$, ***$p \leq 0.001$, ****$p \leq 10^{-4}$, *****$p \leq 10^{-5}$. Real-time qRT- PCR confirming LT-mediated modulation of genes in ileal (Hu235D) enteroids encoding **b** the major thiamine transporter SLC19A3 and **c** the SP1 cis-regulatory element. Data reflect two independent experiments with two replicates each. **d** Uptake of [$^3$H]-thiamine by Hu235D cells is impaired following LT treatment. Data presented in **b**–**d** are from two independent experiments with $n = 3$ replicates each. (*<0.05, **<0.01 by Mann–Whitney two-tailed comparisons).

vaccinated mouse dams with heat-labile toxin and examined the brush border ultrastructure in suckling mice. Vaccinated dams but not sham vaccinated controls expressed significant levels of IgA and IgG in breast milk (Fig. 6a), consistent with increased levels of antibodies in the stomachs of infant mice (Fig. 6b). Notably, we found that maternal vaccination with LT completely abrogated changes to the microvilli (Fig. 6c, d) further substantiating the importance of LT in driving changes to the epithelial architecture.

### LT modulates key transcription factors that govern enterocyte development

Despite the marked alteration in transcription mediated by LT, the majority of genes critical for brush border development lacked conserved CRE sites[23]. Therefore, we performed transcription factor target enrichment analysis[42,43] to identify potential transcription factors responsible for differential regulation of genes significantly ($P < 0.05$) upregulated or downregulated by LT in both Caco-2 and enteroid RNA-seq datasets (Supplementary Table 4 and Supplementary Dataset 1). The upregulated genes were most significantly ($p = 4.2 \times -10^{-3}$) linked to transcription factor targets of AP-1 encoded by the *c-jun* gene, previously shown to be regulated by cAMP[44], and to be involved in intestinal epithelial repair[45]. Notably, however, downregulated genes were most significantly enriched in targets for the HNF4α transcription factor or its intestine-specific paralog HNF4γ ($P = 1.9 \times -10^{-4}$) that were recently shown to regulate multiple genes required for brush border development[46,47]. Importantly, PKA has also been shown to phosphorylate HNF4 α at a consensus recognition site within the DNA binding domain, shared with HNF4 γ, inhibiting transcription[48].

Of note, the transcription of both paralogs was found to be significantly depressed following exposure of intestinal epithelia to LT (Fig. 7a, b), as were levels of HNF4γ in nuclear fractions from toxin-

treated cells (Fig. 7c), suggesting that activation of cAMP can interfere with transcription mediated by HNF4. To further examine the impact of LT on HNF4-mediated transcription, we introduced a transcriptional reporter plasmid containing six tandem copies of the HNF4 transcriptional response element (5′-CAAAGGTCA-3′) linked to a human codon-optimized *Gaussia princeps* luciferase into Caco-2 cells. These assays demonstrated that HNF4-mediated transcription was dramatically reduced in cells treated with LT (Fig. 7d). Similarly, we found that relative to nuclei of intestinal epithelial cells in ileal segments from sham-challenged control mice, those from ETEC-challenged mice exhibited significantly less HNFγ (Fig. 7e) further supporting a role for ETEC in modulating the production of this important transcription factor. Chen et al described a "feed-forward regulatory module" essential to enterocyte differentiation in which HNF4 and SMAD4 transcription factors reciprocally activate each other's transcription. As would be predicted from this model, we found that transcription of SMAD4 was also impaired by LT (Supplementary Fig. 6A, B), leading us to speculate that LT-mediated phosphorylation of HNF4 by PKA, interrupts this critical transcription module (Supplementary Fig. 6C–E).

Collectively, the current studies demonstrate that in addition to their known canonical effects on ion transport that culminate in watery diarrhea, ETEC toxins can drive appreciable derangement of enterocyte architecture and function by interfering with key pathways in intestinal epithelia that govern the formation of mature enterocytes capable of effective nutrient absorption. These findings have important implications for our understanding and prevention of enteropathic conditions linked to ETEC.

## Discussion

Understanding the molecular events that lead to sequelae of undernutrition and growth faltering following ETEC infections may

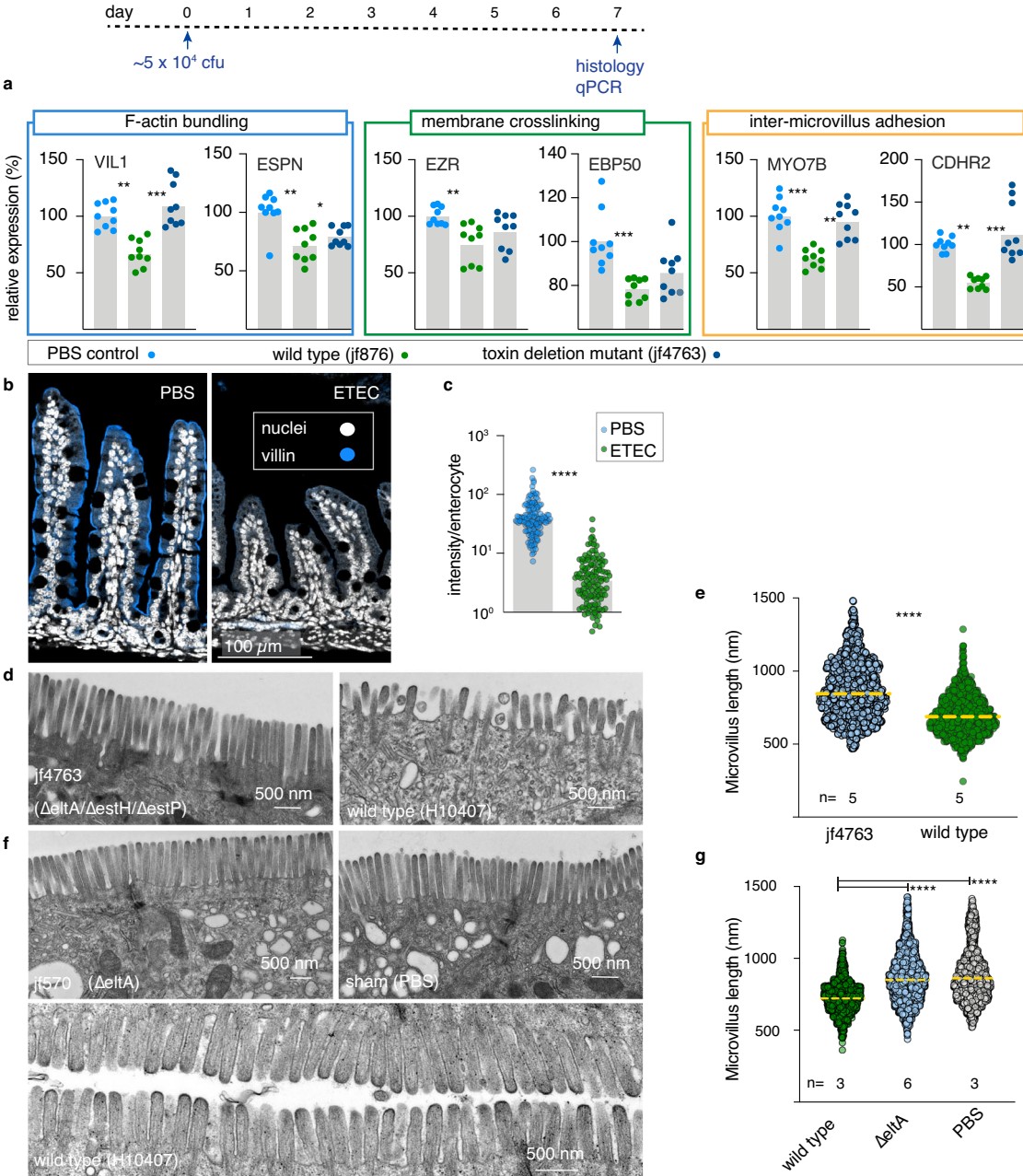

**Fig. 5 | ETEC disrupts in vivo formation of small intestinal microvilli.** Timeline at the top depicts the challenge with ETEC or control nontoxigenic isolate or sham (PBS) challenge. **a** Quantitative PCR results for genes involved in brush border development in small intestinal samples obtained from infant mice (*n* = 9/group) 7 days after challenge with toxigenic ETEC (jf876), nontoxigenic ETEC (jf4763, LT⁻/ST⁻) PBS controls. Comparisons between data represent ANOVA, Kruskal–Wallis testing where ***$p \leq 0.001$, **$p \leq 0.01$, and *$p \leq 0.05$. **b** Immunofluorescence images of small intestinal sections showing villin expression (blue), and nuclei (white). **c** Mean villin fluorescence intensity normalized per enterocyte (*n* = 4 mice/group)

****$p \leq 0.0001$ by Mann–Whitney (two-tailed) nonparametric comparisons. **d** Representative transmission electron microscopy (TEM) images of the small intestinal brush border from mice challenged with toxin-negative (Δ, left) and toxigenic ETEC (wt, right). **e** Microvillus length ****$p \leq 0.0001$ by Mann–Whitney (two-tailed) nonparametric comparisons. Data represent geometric mean length from n = 5 mice per group in three independent experiments. **f** TEM images from mice challenged with jf570 (*eltA*::Km^R), sham PBS controls, or mice challenged with wild-type ETEC. **g** Length of microvilli (dashed horizontal lines represent geometric means). ****$p < 0.0001$ by Kruskal–Wallis.

be key to the effective design of prevention strategies, including vaccines[49]. Although the molecular mechanisms involved in the fluid and ion fluxes into the intestinal lumen leading to diarrhea are firmly established, only recently has evidence emerged to suggest that ETEC toxins may incite previously unappreciated changes in small intestinal epithelia[50,51]. The present studies, initiated to identify additional effects of LT, were prompted by an appreciation that cyclic nucleotides, particularly cAMP, govern a multitude of cellular pathways, potentially resulting in collateral impacts that extend

beyond the acute episodes of diarrhea. Consistent with this model, we found that exposure of intestinal cells to heat-labile toxin altered the transcription of hundreds of genes. A central theme highlighted in the analysis of these transcriptional alterations is that LT affects major classes of genes involved in the biogenesis of microvilli and the function of the intestinal brush border, the major site of nutrient uptake in the small intestine, potentially offering a direct molecular link to sequelae of malnutrition and impaired growth in children.

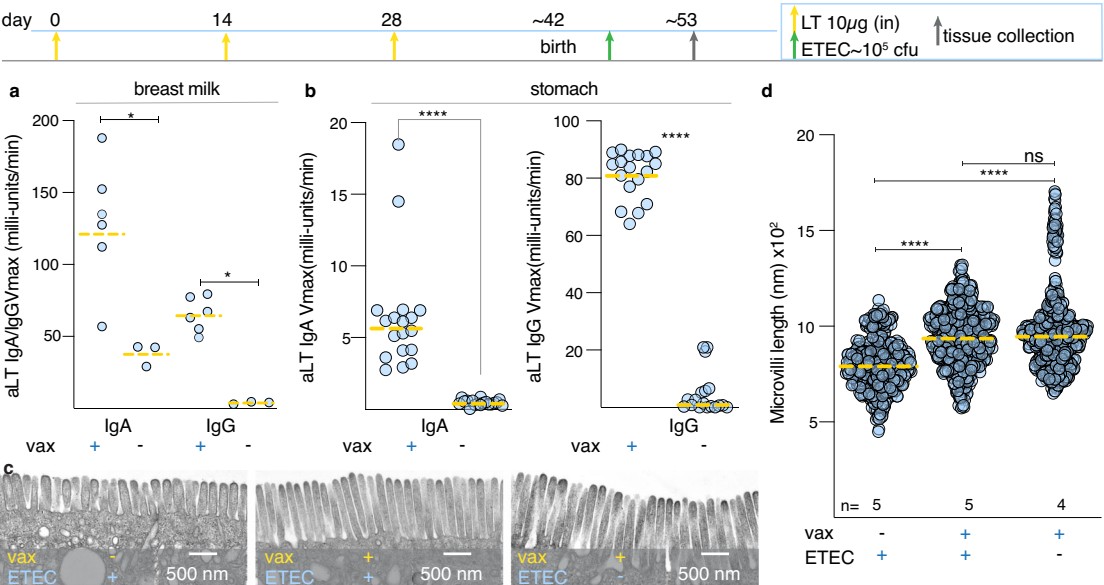

Fig. 6 | **Maternal vaccination with LT mitigates microvillus disruption in neonatal mice.** Timeline depicts vaccination and challenge (top): maternal intranasal (i.n.) vaccinations with 10 µg LT/immunization (yellow arrows on days 0, 14, 28) and neonatal challenge at 3 days of age (green arrow) followed by sacrifice and tissue collection at 7 days post-infection (gray arrow). **a** Kinetic ELISA data of from triplicate samples of breast milk anti-LT (IgA, and IgG in $n = 2$ immunized dams and 1 un-immunized control). *<0.05 by Mann–Whitney two-tailed nonparametric testing. **b** Anti-LT antibodies in the gastric contents of neonatal mice at day 53. ****$p < 0.0001$ by Mann–Whitney two-tailed comparisons. **c** Representative transmission electron microscopy images of brush border microvilli from unvaccinated mice challenged with wild-type ETEC (left), vaccinated mice challenged with wild-type ETEC, and vaccinated un-challenged controls. **d** Microvillus lengths (based on image analysis of $n = 5$ mice group) ****<0.0001 Kruskal–Wallis comparisons.

The potential clinical relevance of the observations reported here is highlighted by remarkably similar ultrastructural alteration of intestinal epithelial cells seen in small intestinal biopsies of patients with acute cholera[25] and tropical sprue[17]. In both entities, the brush border was noted to be abnormal, with shortened, irregular microvilli. Importantly, however, despite the structural and functional similarity between LT and CT, clinical cholera, unlike ETEC infections, has not been linked to enteropathy or attendant sequelae. Whether this relates to the repetitive nature of ETEC infections compared to the durable protective immunity that follows a single *V. cholerae* infection[52] is not presently clear. Similarly, while tropical sprue remains a leading cause of malabsorption in regions where infectious diarrhea is prevalent[53–55], the most debilitating forms of this illness have not typically followed isolated cases of traveler's diarrhea, but occur in resident populations or expatriates[56] repeatedly assailed by diarrhea while residing in endemic regions[57]. Likewise, our data also highlight the potential importance of repeated infections on the development of sequelae.

The negative impacts of LT on brush border architecture, with a commensurate reduction in surface area available for nutrient absorption, are compounded by the alteration of multiple SLC genes that encode transporters critical for the uptake of essential vitamins and other molecules. Importantly, small intestinal biopsies obtained from Zambian children with enteropathy and refractory stunting exhibited similar changes in SLC gene expression profiles[58]. The decreased transcription of molecules required for intestinal zinc uptake, in cells treated with LT is particularly intriguing given the known aberrations in zinc absorption in children with enteropathy[35,59,60], the possible contribution of zinc deficiency to enteropathic changes[61], and the salutary effects of zinc in the treatment of children with diarrhea[62,63].

Further study will be needed to precisely delineate the role of ETEC LT and ST enterotoxins, alone and in combination, in driving enteropathic changes to the intestine and sequelae. ST-producing ETEC were most strongly associated with moderate to severe diarrhea in GEMS[3], and follow-on studies of children enrolled in these studies

have also linked infections with ETEC encoding heat-stable toxin to growth faltering[7]. However, our earlier analysis of a global collection of more than 1100 ETEC isolates, including those collected in GEMS, demonstrated that slightly more than half of all ST-encoding strains also encoded LT, and that roughly one-third of the isolates overall encoded LT alone, ST-LT, or ST only[64]. While LT-producing ETEC have been specifically linked to malnutrition among children in Bangladesh[27], and our in vitro and animal studies point to the potential importance of LT, additional effort will be needed to correlate toxin-induced morphologic and functional perturbation of the intestinal brush border with outcomes in children. Importantly, the long-term morbidity associated with ETEC infections does not appear to correlate with the severity of diarrhea as both mild illness[4], and perhaps asymptomatic colonization may lead to growth faltering.

Further refinement of animal models that can faithfully recapitulate features of enteropathy are also needed. Indeed, conventional mice lack genes that could be required to reproduce the full effects of ETEC. For instance, each of the carcinoembryonic antigen cell adhesion molecules (CEACAMs) that are substantially upregulated on human small intestinal epithelia in response to LT, and which we have recently shown to play a critical role in ETEC interactions with human small intestine[50], are absent in mice.

While the precise mechanism underlying LT-mediated modulation of genes required for microvillus biogenesis and absorptive function of the brush border is presently unclear, stimulation of adenylate cyclase invokes many cAMP-responsive nuclear factors that may serve either to activate or repress transcription. Genes implicated in the development of microvilli are mostly devoid of consensus palindromic (TGACGTCA) or "half" (CGTCA) cAMP-response element (CRE) sites within their promoter regions for direct modulation by CREB[65], which typically is involved as a transcriptional activator, and both CREB and CREM can yield several alternatively spliced variants that may act as either activators or repressors[66]. cAMP second messaging also engages multiple signaling pathways converging at CREB[67], and PKA can phosphorylate and modulate the activity of multiple

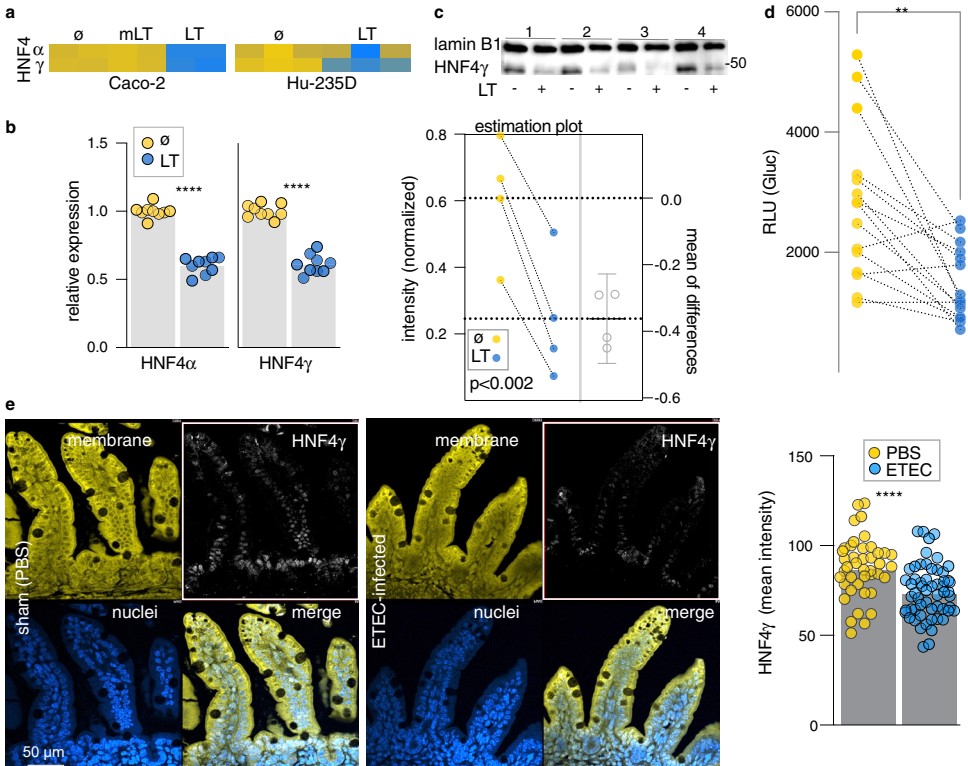

**Fig. 7 | Enterotoxigenic *E. coli* heat-labile toxin impairs production of HNF4 nuclear receptors. a** Heatmap demonstrating the impact of LT on transcription of paralogous transcription factors HNF4α and HNF4γ in Caco-2 cells (left) and ileal enteroids, (Hu235D, right) ø untreated, mLT mutant LT. **b** qRT-PCR (TaqMan) data confirming decreased transcription of HNF4 transcription factors following treatment of enteroids with LT (*n* = 9 biologically independent samples). ****<0.0001 by Mann–Whitney two-tailed comparisons. **c** HNF4γ is decreased in nuclear fractions obtained from small intestinal enteroids following treatment with LT. Shown in the HNF4γ immunoblot are samples from four independent experiments, with the graph below-representing quantitation of signal intensity normalized to the lamin B1 nuclear protein (*p* = 0.0017, paired *t*-test, one-tailed). Bars indicate mean ± 95% confidence intervals. **d** HNF4 transcription *Gaussia* luciferase reporter assay showing a decrease in signal following treatment of TR104-transfected Caco-2 cells with LT. 2 experimental replicates (*n* = 15 samples total) ***p* < 0.001 Wilcoxon matched pairs, one-tailed). **e** Confocal microscopy of representative Ileal sections from sham-challenged (PBS) left, and ETEC-infected mice (right). Immunofluorescence intensity of HNF4γ signal in sections from *n* = 5 control mice, and *n* = 6 ETEC-challenged mice. Membranes (yellow) were strained with CellMask orange (Thermo Fisher C10045), nuclei (blue) were stained with DAPI, and HNFγ immunostaining was represented in white. Each symbol represents a microscopic region of interest. Bars represent geographic means (*p* < 0.0001, Mann–Whitney two-tailed comparisons).

transcription factors to act either as transcriptional activators or repressors, including SP1[68]. Notably, putative binding sites for HNF4, a cAMP-modulated transcription factor[48] known to regulate genes needed for the formation of microvilli[46,47], were significantly enriched in the promotors of genes downregulated by LT. Both HNF4α and HNF4γ possess canonical PKA recognition sites within their DNA binding motifs, and PKA phosphorylation of these sites interrupts transcription[69]. HNF4 activates the transcription of SMAD4, and in turn, SMAD4 activates the transcription of HNF4[70]. Both transcription factors then engage genes needed for the effective differentiation of stem cells to mature enterocytes[70]. Modulation of the activity of this transcription factor by LT would therefore be predicted to have a marked impact on pathways critical to intestinal epithelial homeostasis.

We should also note that increases in cellular cAMP can impact multiple cellular pathways independent of PKA. Included among these are pathways governed by a more recently discovered family of cellular cAMP-binding molecules, exchange proteins activated by cAMP (EPACs). EPACs appear to play critical roles as guanine exchange factors that regulate GTPase proteins[71] and are involved in complex signaling networks implicated in cell growth, differentiation, and morphogenesis[72,73].

cAMP can also exert potent epigenetic influences on transcription. CREB-binding protein (CBP) possesses intrinsic histone acetyltransferase (HAT) activity[74], and can therefore modulate chromatin

remodeling, enhancing access to transcription factors. In addition, cAMP messaging through PKA leads to phosphorylation-dependent activation of the histone demethylase enzyme PHF2 to promote the transcription of multiple genes that can impact the transition from stem cells to epithelial cells[75].

Increased intestinal permeability is a recognized hallmark of enteropathy in young children in LMICs[76]. Given the known negative impacts of cholera toxin on epithelial barrier function[77], and its structural and functional similarity to a heat-labile toxin, LT may exert additional enteropathic effects beyond the impact on brush border biogenesis described here.

Altogether it seems likely that multiple pathways governed by increases in cellular cAMP may underlie the morphologic and functional disruption of the brush border epithelial observed in our studies. Nevertheless, the data presented here provide compelling evidence that the heat-labile toxin of ETEC ultimately impacts multiple genes required for the biogenesis and function of the brush border, the major site of nutrient absorption in the human small intestine. The findings may have significant implications for our understanding of sequelae linked to ETEC, including environmental enteropathy in young children, and tropical sprue in adults. The increased acknowledgement of long-term morbidity linked to ETEC and an improved understanding of the role of ETEC enterotoxins as drivers of this morbidity may also strengthen the case for vaccines[49] specifically engineered to prevent both acute illness and sequelae.

## Methods

The studies reported here comply with all relevant ethical regulations and the study protocols have been approved by the IACUC, Institutional Biosafety, and Institutional Review Boards at Washington University in Saint Louis School of Medicine.

### Culture and differentiation of enteroids

Archived small intestinal enteroid specimens derived from biopsy samples of adult patients undergoing routine endoscopy were obtained from the Organoids Core of the Digestive Disease Research Core Center (DDRCC) at Washington University School of Medicine under Institutional Review Board (IRB) protocol number 201406083. Cell lines used in these are routinely validated for response to the heat-labile toxin by cAMP assays (Arbor Assays, Ann Arbor, MI).

Purified crypt cells from the ileum (Hu235D) were resuspended in Matrigel (BD Biosciences, San Jose, CA, USA) and 15 μL of resuspended matrix gel was added to each well in 24 well plates. Plates were incubated at 37 °C and 5% $CO_2$ with 50% L-WRN conditioned media (CM) and 50% primary culture medium (Advanced DMEM/F12, Invitrogen) supplemented with 20% FBS, 2 mM L-glutamine, 100 units/mL penicillin, 0.1 mg/mL streptomycin, 10 μM Y-27632 (ROCK inhibitor, Tocris Bioscience, R&D systems, Minneapolis, MN, USA), and 10 μM SB431541 (TGFBR1 inhibitor, Tocris Bioscience, R&D systems).

To induce differentiation and polarization of enteroids, cells were washed once to remove Matrigel, followed by trypsinization and centrifugation at $1100 \times g$ for 5 min. Cells were then resuspended in 1:1 CM and primary medium with Y-27632 and SB431541 as described above. Cells were then plated on semiporous filters (Transwells®, 6.5 mm insert, 24 well plate, 0.4-μm polyester membrane, Corning Incorporated, Kennebunk, ME, USA) that had been previously coated with Collagen IV (Millipore Sigma). Transwell® inserts were rinsed with DMEM/F12 with Hepes, 10% FBS, L-glutamine, Penicillin, and streptomycin. Cells were allowed to grow to confluency in 50% conditioned media (CM) and then changed to a differentiation medium (5% CM in primary medium + ROCK inhibitor) for 48 h. Differentiated cells were used for toxin treatment.

### Propagation of Caco-2 cells

Caco-2 intestinal epithelial cells were obtained from ATCC (ATCC HTB-37) and cultured at 37 °C, 5% $CO_2$, in Eagle's MEM supplemented with 20% of fetal bovine serum (FBS). To generate polarized monolayers, ~$1 \times 10^5$ cells were seeded onto polystyrene membrane filters (0.4 μM, 6.5 mm diameter insert, Transwell, Corning) and grew for at least a week prior to toxin treatment. Media was replaced with fresh media every two days.

### Toxin treatment

Polarized differentiated cells were treated with a heat-labile toxin (LT) in a differentiation medium (100 ng/mL) in a volume of 700 μL at the basolateral side of the Transwell insert and 100 μL volume on the apical aspect of the monolayer. Treated cells were incubated at 37 °C for 18 h and then fixed for transmission electron microscopy or fluorescence microscopy, or lysed for RNA extraction (GE Healthcare, Buckinghamshire, UK). LT and mutant LT (E211K, mLT) were kindly provided by Dr. John D. Clements, Tulane University, New Orleans, Louisianna, USA.

### RNA extraction and cDNA synthesis

RNA was extracted using the Illustra RNAspin Mini RNA extraction kit (GE Healthcare, Buckinghamshire, UK). Three biological replicates per treatment (untreated and LT treated) were submitted to the Genome Technology Access Center (GTAC) at Washington University in St. Louis School of Medicine for RNA-seq library preparation. An Agilent Bioanalyzer was used to determine the integrity of RNA samples. cDNA was generated using Superscript™ Vilo™

cDNA synthesis kit (Invitrogen by Thermo Fisher) after normalizing RNA concentrations.

### RNA-seq library preparation

Library preparation was performed with 10 ng of total RNA, integrity was determined using an Agilent bioanalyzer. ds-cDNA was prepared using the SMARTer Ultra Low RNA kit for Illumina Sequencing (Clontech) per the manufacturer's protocol. cDNA was fragmented using a Covaris E220 sonicator using peak incident power 18, duty factor 20%, cycles/burst 50, and time 120 s. cDNA was blunt-ended, had an A base added to the 3'ends, and then had Illumina sequencing adapters ligated to the ends. Ligated fragments were then amplified for 12–15 cycles using primers incorporating unique index tags. Fragments were sequenced on an Illumina HiSeq-2500 using single reads extending 50 bases.

### RNA-seq data processing and analysis

RNA-seq reads were aligned to the Ensembl top-level human genome assembly with STAR version 2.7.3a. Gene counts were derived from the number of uniquely aligned unambiguous reads by Subread:featureCount version 1.54.1. Read counts were used as input for DESeq2 differential gene expression analysis (version 1.24.0)[78] with default settings, and a minimum $P$ value significance threshold of $10^{-5}$ (after false discovery rate [FDR[79]] correction for the number of tests). Fragments per kilobase per million reads mapped (FPKM) values for relative gene expression were calculated from DESeq2-normalized read counts, and $Z$-scores were calculated per gene using the average and standard deviations of FPKM values across samples. $Log_2$ fold changes were identified from differential expression output. Pathway enrichment analysis for KEGG[80] and Gene Ontology (GO)[81] pathways among gene sets of interest was performed using the over-representation analysis tool provided on the WebGestalt[42] web server (version 2019). Heatmap and bar graph visualization was performed with Microsoft Excel.

Transcript counts were produced by Sailfish version 0.6.3. Sequencing performance was assessed for the total number of aligned reads, the total number of uniquely aligned reads, genes and transcripts detected, ribosomal fraction known junction saturation, and read distribution over known gene models with RSeQC version 2.3.

All gene-level and transcript counts were then imported into the R/Bioconductor package EdgeR and TMM normalization size factors were calculated to adjust for samples for differences in library size. Ribosomal features, as well as any feature not expressed in at least the smallest condition size minus one sample were excluded from further analysis and TMM size factors were recalculated to created effective TMM size factors. The TMM size factors and the matrix of counts were then imported into R/Bioconductor package Limma and weighted likelihoods based on the observed mean-variance relationship of every gene/transcript and sample were then calculated for all samples with the voomWithQualityWeights function. The performance of the samples was assessed with a spearman correlation matrix and multidimensional scaling plots. Gene/transcript performance was assessed with plots of the residual standard deviation of every gene to their average log count with a robustly fitted trend line of the residuals. Generalized linear models were then created to test for gene/transcript level differential expression. Differentially expressed genes and transcripts were then filtered for FDR-adjusted $p$ values less than or equal to 0.05.

The biological interpretation of the large set of features found in the Limma results were then elucidated for global transcriptomic changes in known Gene Ontology (GO) and KEGG terms with the R/Bioconductor packages GAGE and Pathview. Briefly, GAGE measures for perturbations in GO or KEGG terms based on changes in observed $log_2$ fold changes for the genes within that term versus the background $log_2$ fold changes observed across features not contained in the

respective term as reported by Limma. For GO terms with an adjusted statistical significance of FDR ≤0.05, heatmaps were automatically generated for each respective term to show how genes co-vary or co-express across the term in relation to a given biological process or molecular function. In the case of KEGG-curated signaling and metabolism pathways, Pathview was used to generate annotated pathway maps of any perturbed pathway with an unadjusted statistical significance of $p$ value ≤0.05. Genes significantly modulated in both Caco-2 cells and enteroids were subjected to further ontology-free analysis via CompBio v1.4 (PercayAI, Inc., www.percayai.com/compbio)[82] to identify unifying biological themes in sets of genes differentially expressed between pairwise comparisons of different groups.

## Quantitative real-time PCR

Quantitative real-time PCR was performed using a QuantStudio 3 real-time detection system (Applied Biosystems). Fast SYBR Green master mix (Applied Biosystems/Thermo Fisher) was used for qPCR analysis. Disassociation curve analysis was performed to assess the specificity of amplification for each sample, and PCR product size was verified by agarose gel electrophoresis. Percent expression was normalized by GAPDH and analyzed using the comparative threshold cycle (Ct) method. Amplification of SLC19A3 transcripts from small intestinal enteroids was performed as recently described[83] with relative gene expression normalized to β actin. HNF4a, HNFg, and SMAD4 gene expression was determined by TaqMan (Thermo Fisher) using validated primer and probe sets. Primers and TaqMan probes used in this study are listed in Supplemental Table 2.

## Transcriptional reporter assays

To assess the impact of LT on HNF4-mediated transcription, pTR104 (GeneCopoeia) carrying six copies of the HNF4 transcriptional response element upstream of a secreted *Gaussia* luciferase (Gluc) reporter gene was transfected (FuGENE, Promega) into confluent Caco-2 cells. Media (EMEM, ATCC) were exchanged 24 h after transfection, and after an additional 24 h, baseline samples of media were removed from each well and stored at −80 °C. Wells were then treated with a heat-labile toxin (100 ng/ml) overnight (16 h). At the end of treatment cell culture supernatants media were transferred to black-walled microplates (Greiner 655086) and processed as directed (Secrete-Pair Luminescence, GeneCopoeia, Inc. Rockville, MD), followed by luminescence detection (Synergy H1, BioTek). Data were expressed as Relative Light Units (RLU) before and after treatment with LT.

SMAD4-mediated transcription was assessed by transient transfection of Caco-2 cells with the SBE4-luc plasmid containing four copies of the SMAD4 transcriptional response element 5′-GTCTAGAC-3′[84].

Following treatment with overnight treatment with LT, cells were resuspended in 80 μl of EMEM media and mixed with an equal volume of ONEGLO EX Reagent (Nano-Glo® Dual-Luciferase® Reporter Assay System, Promega) for 20 min on an orbital shaker at 300 rpm, then read on the luminometer as above.

## Immunoblotting

**SLC19A3, SP1.** Total protein was extracted from LT toxin-treated (100 ng/ml) human differentiated enteroid monolayers (235D line) and untreated control using radioimmunoprecipitation assay (RIPA) buffer (Sigma) containing protease inhibitor cocktail. An equal amount (~25 μg) of the proteins were loaded on a NuPAGE 4-12% Bris–Tris gradient gels (Invitrogen) as previously described[83], then blotted onto polyvinylidene difluoride (PVDF) membranes and probed with anti-SLC19A3 (1:1000; Cat# 13407-1-AP; Proteintech), or anti-SP1 (1:1000; Cat# ab124804; Abcam) antibodies and together with anti-beta actin (1:3000; Cat# sc-47778) primary antibodies. The specificity of the SLC19A3 antibodies was validated in our laboratory previously using different approaches that include overexpression of tagged protein or gene silencing[85]. Anti-SP1 antibodies were validated by the manufacturer using knockout cell lysate protein samples. The SLC19A3 and SP1 protein bands from the blot were then identified with corresponding anti-rabbit IR-800 dye (1:30,000; Cat# 926-32211; LI-COR Bioscience) and anti-mouse IR-680 dye (1:30,000; Cat# 926-68020; LI-COR Bioscience) secondary antibodies incubation for 1 h at room temperature. Relative expression of specific proteins was calculated by comparing the fluorescence intensities in an Odyssey infrared imaging system (LI-COR Bioscience) with respect to the corresponding beta-actin signal.

**Villin.** Differentiated LT-treated and control enteroid monolayers were lysed using NE-PER™ nuclear and cytoplasmic extraction reagents (Thermo Scientific). Cell membrane pellets were solubilized in PBS containing 1% Triton X-100 supplemented with a protease inhibitor cocktail (Pierce™ protease inhibitor mini, Thermo Scientific). Equal amounts of total protein were loaded on a 4–20% gradient SDS-PAGE gel (Mini Protean TGX, Bio-Rad), then blotted onto nitrocellulose membranes and probed with anti-villin mouse monoclonal antibody (1:1000; Cat# SC-66022; Santa Cruz Biotechnology) followed by detection with HRP-conjugated anti-mouse secondary antibody (1:1000; Cat# 7076 S; Cell Signaling Technology). Blots were developed with Clarity Western ECL substrate (Bio-Rad) and imaged with a c600 imaging system (Azure Biosystems). Relative Villin expression signals were then analyzed with respect to corresponding Coomassie-stained gel using ImageJ analysis software.

## Thiamin uptake

Initial rates (30 min; at 37 °C) of carrier-mediated thiamin uptake were examined in LT-treated (100 ng/ml; overnight) and untreated control differentiated small intestinal enteroid monolayers incubated in Krebs−Ringer buffer (pH 7.4) containing [3H]-thiamin (15 nM). Enteroid monolayers were then washed with ice-cold Krebs−Ringer buffer, followed by lysis with NaOH and neutralization with 10 N HCl. The radioactive content was counted using a liquid scintillation counter as described previously[83]. Uptake of thiamin by its respective and distinct carrier-mediated mechanism was determined by subtracting uptake of [3H]-thiamin in the presence of a high pharmacological concentration (1 mM) of unlabeled thiamin from uptake in their absence; all uptake data points were calculated relative to total protein content (in milligrams) of the different preparations and presented as a percentage relative to simultaneously performed controls.

## Confocal microscopy

Cell monolayers were fixed with 2% paraformaldehyde for 30 min at 37 °C and then washed 3× with PBS prior to blocking with 1% BSA in PBS for 1 h at room temperature. Villin expression was detected using 1:100 dilution of anti-villin antibodies raised in mice (Santa Cruz Biotechnology) for 1 h at room temperature, followed by additional 1 h incubation with 1:200 dilution of fluorescence-tagged goat anti-mouse IgG Alexa Fluor 488 (Thermo Fisher) (Supplemental Table 5). Cell membranes were stained with 1:2000 dilution of CellMask orange (Invitrogen) and nuclei were counterstained with DAPI (1:1000) and mounted on glass slides using Prolong Gold antifade reagent (Invitrogen). Images were captured and analyzed on a Nikon C2 confocal microscope equipped with NIS-Elements AR 5.11.01 software (Nikon).

## Electron microscopy

For ultrastructural analyses, in vitro-grown differentiated polarized monolayers of human ileal enteroid samples, as well as mouse intestinal biopsy samples, were fixed in 2% paraformaldehyde/2.5% glutaraldehyde (Ted Pella, Inc., Redding, CA) in 100 mM sodium cacodylate buffer, pH 7.2 for 2 h at room temperature and then overnight at 4 °C. Samples were washed in sodium cacodylate buffer and postfixed in 2%

osmium tetroxide (Ted Pella, Inc) 1 h at room temperature. Samples were then rinsed in $dH_2O$, dehydrated in a graded series of ethanol, and embedded in Eponate 12 resin (Ted Pella, Inc.). Sections of 95 nm were cut with a Leica Ultracut UCT ultramicrotome (Leica Microsystems Inc., Bannockburn, IL), stained with uranyl acetate and lead citrate, and viewed on a JEOL 1200 EX transmission electron microscope (JEOL USA Inc., Peabody, MA) equipped with an AMT 8 megapixel digital camera and AMT Image Capture Engine V602 software (Advanced Microscopy Techniques, Woburn, MA). Images were analyzed using ImageJ software for microvilli length and structures.

### Identification of potential transcription factor target sites

The CREB targets database[65] (http://natural.salk.edu/CREB/) was used to identify potential categorical CRE full (TGACGTCA) or half (TGACG/CGTCA) sites within the promotor regions of genes modulated by the heat-labile toxin. Following previously described protocol[23], 5 kilobase upstream sequences from each human gene were retrieved from Ensembl[86] and were searched for the CREB-binding sites TGACGTCA and TGACGTAA (and their reverse complements), based on sequences retrieved from the TRANSFAC[87] database (matrix ID: V$CREB_01). CREB sequence matches were identified in the upstream sequences of 1,302 genes. WebGestalt[42] was used for overall functional enrichment of potential transcription factor binding sequences upstream from genes that were significantly ($p < 0.05$) up- and down-regulated in both Caco-2 cells and enteroids.

### Suckling mouse challenge studies

Studies in mice were performed under protocol 20-0438 approved by the Washington University in Saint Louis Institutional Animal Care and Use Committee. In neonatal mouse challenge studies, 3-day-old CD-1 mice (male and female) were inoculated with wild-type ETEC (H10407), isogenic toxin deletion mutants, or sterile PBS (sham negative controls). Inocula were prepared from frozen glycerol stocks maintained at −80 °C grown overnight (-16 h) in 2 ml Luria Bertani (LB) broth in a shaking incubator (at 37 °C, 225 rpm). Cultures were then diluted 1:100 in 20 ml of fresh LB, grown for an additional 2 h, and bacteria were harvested by centrifugation (6000 rpm, 5 min at 4 °C). Pellets were then washed once in ice-cold sterile PBS, then resuspended to a density of ~$1 \times 10^5$ colony forming units (CFU) per 20 µl inoculum. CFU in each inoculum were determined by plating serial dilutions LB-agar plates. Mice were then inoculated with ETEC directly into the stomach through the abdominal wall using a 31 g insulin needle. Following infection, mice were marked with tattoo ink to define groups and returned to the cage. Seven days post-infection, mice were sacrificed, and the ileal tissue collected was fixed for microscopy or preserved for RNA extraction.

### Bacterial strain construction

The strains used in these studies are described in Supplementary Table 1. Strain jf4763 is devoid of all enterotoxins and was constructed from a previously engineered ST-negative mutant of H10407, jf2848[88]. Briefly, jf2848 was first transformed with pKD46, a lambda red recombinase helper plasmid with selection on ampicillin. Jf2848(pKD46) was then transformed with a PCR amplicon generated with primers _to replace the *eltAB* genes encoding LT with a spectinomycin resistance cassette resulting in jf4763. The strain was subsequently validated by PCR and by testing supernatants in GM1 ganglioside ELISA to confirm the lack of LT production.

### Vaccination and immunologic assessment

Adult female mouse dams were vaccinated intranasally three times at 2-week intervals with 10 µg LT holotoxin in PBS, with the last dose administered ~2 weeks prior to anticipated delivery. Control mice were mock vaccinated with PBS. Fecal pellets collected from vaccinated dams were used to assess mucosal responses to LT. To collect breast milk, the dam was separated from the litter ~2 h prior to milking. Oxytocin (Sigma O3251), 0.002 IU/g, was injected IP to stimulate milk production, followed by collection via pipette tip. Anti-LT immune responses in maternal fecal extracts and breast milk, and stomach contents and sera of neonates (male and female) were assessed by GM1 ganglioside[89] kinetic ELISA[90] for anti-LT IgA and IgG, using plates coated with GM1 (Sigma G7641), as previously described[91].

### Reporting summary

Further information on research design is available in the Nature Portfolio Reporting Summary linked to this article.

## Data availability

RNAseq data were deposited at the Sequence Read Archive (SRA) on the NCBI website https://www.ncbi.nlm.nih.gov/sra under BioProject accession number PRJNA875141. Primary source data and original images are available on Figshare as detailed in Supplementary Table 6 of the supplementary information.

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

## Acknowledgements

J.M.F. was supported by funding from the National Institute of Allergy and Infectious Diseases (NIAID) of the National Institutes of Health (NIH) R01 AI126887, R01 AI089894, U01 AI095473; and the Department of Veterans Affairs (5I01BX001469-05). Research conducted by AS was

also supported by the National Institute of Allergy and Infectious Diseases of the National Institutes of Health under Award Number T32AI007172. We thank the Genome Technology Access Center in the Department of Genetics at Washington University School of Medicine for help with genomic analysis. The Center is partially supported by NCI Cancer Center Support Grant #P30 CA91842 to the Siteman Cancer Center and by ICTS/CTSA Grant# UL1TR002345 from the National Center for Research Resources (NCRR), a component of the National Institutes of Health (NIH), and NIH Roadmap for Medical Research. H.M.S. was supported by funding from the Department of Veterans Affairs (I01BX001142) and the National Institutes of Health (DK5606 and AA018071). HMS is the recipient of a Senior Research Career Scientist award # IK6BX006189.

The content is solely the responsibility of the authors and does not necessarily represent the official views of the National Institutes of Health, or the Department of Veterans Affairs.

## Author contributions

A.S., B.T., T.J.V., S.S., R.D.S., and W.B., performed experiments. J.C.M., B.A.R., M.M., C.S. E.T., and R.D.H., analyzed data. H.M.S., M.M., and J.M.F. planned experiments and wrote the manuscript

## Competing interests

R.D.H. and C.S. may receive royalty income based on the CompBio technology developed by R.D.H. and licensed by Washington University to PercayAI. The remaining authors declare no competing interests.

## Additional information

Alaullah Sheikh [ID][1], Brunda Tumala[1], Tim J. Vickers[1], John C. Martin[2], Bruce A. Rosa [ID][1,2], Subrata Sabui[3,4], Supratim Basu [ID][1], Rita D. Simoes[1], Makedonka Mitreva [ID][1,2], Chad Storer[2,5], Erik Tyksen [ID][2,5], Richard D. Head[2,5], Wandy Beatty[6], Hamid M. Said[3,4] & James M. Fleckenstein [ID][1,7] ✉

[1]Division of Infectious Diseases, Department of Medicine, Washington University School of Medicine, St. Louis, MO 63110, USA. [2]The McDonnell Genome Institute, Washington University School of Medicine, St. Louis, MO 63110, USA. [3]Departments of Medicine and Physiology/Biophysics, School of Medicine, University of California-Irvine, Irvine, CA 92697, USA. [4]Department of Research, VA Medical Center, Long Beach, CA 90822, USA. [5]Department of Genetics, Washington University School of Medicine, St. Louis, MO 63110, USA. [6]Department of Molecular Microbiology, Washington University School of Medicine, Saint Louis, MO 63110, USA. [7]Infectious Diseases, Medicine Service, Veterans Affairs Saint Louis Health Care System, Saint Louis, MO 63106, USA. ✉e-mail: jfleckenstein@wustl.edu

