## [Peer Review File · Nature Communications]

Enterotoxigenic *Escherichia coli* heat-labile toxin drives enteropathic changes in small intestinal epitheliaREVIEWER COMMENTS

Reviewer #1 (Remarks to the Author):

In this interesting paper, the authors establish that ETEC heat labile toxin (LT) - which is well studied in terms of its ability to stimulate pathogenic epithelial fluid secretion - also induces profound perturbations to brush border structure and the expression of key proteins that normally reside in the enterocyte apical domain. This study combines RNA seq analysis with tissue section staining and ultrastructural imaging to investigate how LT impacts human caco-2 cells, human enteroids, and mouse intestinal tissues in vivo. All of the data are consistent with the conclusion that LT significantly disrupts apical structure and function, most likely by impacting the expression of brush border protein machinery. In general, the paper is clearly written and the conclusions made by the authors are supported by the data presented in the main and supplemental figures. These results are important because they identify a new dimension to LT scope of action, which might explain some of the sequelae to ETEC exposure, including enteropathy and nutrient malabsorption. Below I list a few points that I hope the authors will consider toward improving the manuscript.

Figure 1, The structure showing the mutation is of limited value to the reader here unless other functionally relevant motifs in LT are identified and labeled. Also, perhaps Figure 1 could be combined with figure 2 as thematically they are highly related.

Figure 2, The color coding in the cartoon and in the expression data is a confusing, as yellow is used to denote both microvillar adhesion factors as well as higher expression?

Figure 3, Control villin staining in B panel does not look like one would expect; the yellow signal appears throughout the entire cell rather than the usual enrichment at the apical surface. This point also calls into question the complete lack of villin signal shown in the LT treated cells. Moreover, a complete lack of villin signal does not line up with the fold-change in villin intensity data shown in the plots of the right. The authors need to revisit this analysis, and perhaps stain for additional brush border structural proteins identified in their RNAseq data, to provide more points of validation.

Figure 3, In this and other figures microvillus length is shown with units as $\text{nm} \times 10^3$? The exponential notation here is needless and confusing for the reader. Just change the units to μm .

Figure 5, what is the difference between the normalization shown in panel C versus panel D? These seem redundant.

In more than one figure, the intensity of the villin signal is expressed relative to the DAPI nuclear stain. Unfortunately, in sectioned intestinal tissue, the nucleus might not be captured/well represented in the image, which in turn would impact the denominator in this normalization scheme. Because villin signal is normally apically enriched, another well accepted approach would involve normalizing the apical signal relative to the cytoplasmic or even whole cell levels. In this case, the ETEC treated ratio would be closer to 1 whereas the control would be much higher than 1.

Figure legends ask the reader to view the individual panels in a series of images e.g. "clockwise from left" - this is confusing and simply not clear. When showing a series of images, please label each image with a letter and refer to that letter in the figure legend.

Reviewer #2 (Remarks to the Author):

The focus of this manuscript is the impact of enterotoxigenic *Escherichia coli* (ETEC) heat-Labile Toxin (LT) on intestinal epithelial cells. The authors demonstrate that LT has broad impacts on the transcriptomes of Caco-2 cells and human ileal enteroids. Consistent with its ability to decrease expression of genes involved in microvilli biogenesis, LT caused shortening and distortion of microvilli in enteroids, and in neonatal mouse intestines. Maternal vaccination with LT blocked

intestinal brush border disruption in ETEC-challenged neonatal mice.

Overall, the approach and methodology are sound, and the observations are noteworthy and novel. The findings add to our knowledge of LT-induced alterations of the intestinal mucosa. The following are some recommendations to improve the manuscript and bolster the findings.

1. The manuscript primarily focuses on alterations in RNA abundance (RNAseq, qRT-PCR), with correlative protein data only for villin (confocal images) and HNF4 γ (Western blots, confocal). The findings will be considerably strengthened by western blot data for additional proteins (brush-border components, SLC19A3, SP1, SMAD4).
2. What was the rationale for the chosen LT concentration/treatment duration (100 ng/ml for 18 hrs)? LT (50ng/ml for 16 hrs) was previously reported to induce >40% apoptosis of Caco-2 cells (Lu et al, 2017; <https://doi.org/10.3389/fcimb.2017.00244>). Was there significant host cell death with the treatment condition used in the current study at 18 hours (or 72 hours in Fig. 3)?
3. Lines 488 & 504: Please indicate if both male and female pups/neonates were included in the study.
4. Figure 1 legend: "heat-labile toxin relative to untreated cells or cells treated with the biologically inactive mLT". The use of "or" in this sentence is confusing: is this intended to be 3-way comparison (LT vs. mLT vs. untreated)? Also, the X-axis label "untreated/mLT" seems to suggest a ratio. Is this the intention? Should the first line of the X axis labels be identical for 1A and 1B? Also, please specify the comparator (differential gene expression/higher/lower relative to ...?) in Supplemental Table 3.
5. Figure 2B (also Fig. 4, Fig 7A): For replicates, it appears that n = 2 for each treatment for Caco-2 cells (sham, mLT and LT), and n = 3 for ileal enteroids for the RNAseq data. Is this correct? Please specify in figure legend as appropriate. Also, were enteroids also treated with mLT? If so, include this data in the figure for consistency. It would be instructive to know if the decreases in ESPN and EBP50 are also seen with mLT treatment (as in mouse tissues in Fig. 5; please see next comment).
6. Figure 3: Line 114-115 indicates that the inactive toxin failed to induce significant changes in the transcriptome of Caco-2 cells, but it appears that the inactive toxin was almost as effective as the active molecule (please provide statistics for mLT vs. PBS control) in decreasing the expression of ESPN and EBP50. This should be addressed.
7. Figure 5: (5A) What was the rationale for using a lacZYA::Kmr derivative instead of the parent H10407? (5G) jf570 is not listed in Supplemental table 1 – should it be jf571? It is stated in Supplemental Table 1 that jf4763 was generated as part of the current study. If so, please provide details about strain construction in the Methods section.
8. Figure 6: Please include data on bacterial burden in stool/tissues for all animals. It would be informative to know if the pups were protected against microvillus disruption despite comparable bacterial burdens (or if, somehow, maternal LT vaccination also curtailed ETEC colonization/persistence).
9. Supplemental Figure 4: This model is speculative and is not adequately supported by the data provided. Some experimental validation is warranted: Are LT-mediated transcriptional changes (or brush border alterations) reversed in transfected cells expressing HNF4 and/or SMAD4? Does PKA inhibition inhibit LT-mediated changes in HNF4/SMAD4 and downstream targets? Is there an increase in phosphorylated HNF4 in LT-treated cells?

Minor issues:

1. The arbitrary use of small case letters at the beginning of sub-headings etc. is distracting. Please adhere to standard English conventions throughout the document.
2. Where possible, please include toxin concentration and treatment duration in the figure legends.
3. Line 114-115: For precision, please delete "binding of".
4. Line 192 – end of Results section: This reverts to data on enteroids/Caco-2 cells. For better cohesion, please consider moving this section before the mouse data (Line 163).
5. Line 352: Please indicate LT source
6. Figure 3 legend: Please specify cell type, LT concentration and treatment duration, number of replicates and number of fields assessed.
7. Figure 6: In the schematic, distinguish between maternal treatments (LT) and infant mouse parameters (ETEC infection and tissue collection) - possibly with downward arrows for the former.

Reviewer #3 (Remarks to the Author):

This manuscript provides compelling data that heat-labile toxin (LT) from ETEC is capable of reducing expression of several genes involved in nutrient transport. Functional confirmation of reduced expression of the major thiamine transporter is provided by showing reduced transport of the micronutrient thiamine (3H-thiamine). The authors use complementary Caco-2 IEC lines, primary human intestinal enteroid cultures and in vivo infection models. Mechanistically, the paper implicates a cAMP-PKA mediated suppression of HNF4 target genes involved in microvilli development and transporter gene expression. The paper is well written although the data do not achieve the stated goal of functionally connecting LT infection with ETEC induced enteropathic disease.

The quantification graph for Fig 3B should mention villin to make it easier for the reader to identify the relevance of the figure relative to the immunostaining images in panel 3B.

Did the authors confirm that LT treatment did not increase tight junction permeability as this would negatively impact upon epithelial monolayer integrity and transporting function?

In the in vivo mouse challenge model, a 7-day infection with the LT/ST competent wild-type ETEC induced reduced expression of nutrient transporters and altered microvillus architecture, consistent with the in vitro IEC challenge studies. However, this had no effect on the growth kinetics of the mice, suggesting that ETEC did not compromise development. Seven days may be too short a timeframe to identify an effect on weight loss. In order for the authors to support their hypothesis that ETEC infection can predispose to non-diarrheal and enteropathic pathologies, some evidence of a pathologic event arising from microvillus shortening and/or nutrient deprivation needs to be demonstrated. The absence of such evidence undermines the translational relevance of the study which is the stated major emphasis of this paper.

Point-by-point response to reviewer critiques

Reviewer #1 (Remarks to the Author):

In this interesting paper, the authors establish that ETEC heat labile toxin (LT) - which is well studied in terms of its ability to stimulate pathogenic epithelial fluid secretion - also induces profound perturbations to brush border structure and the expression of key proteins that normally reside in the enterocyte apical domain. This study combines RNA seq analysis with tissue section staining and ultrastructural imaging to investigate how LT impacts human caco-2 cells, human enteroids, and mouse intestinal tissues *in vivo*. All of the data are consistent with the conclusion that LT significantly disrupts apical structure and function, most likely by impacting the expression of brush border protein machinery. In general, the paper is clearly written and the conclusions made by the authors are supported by the data presented in the main and supplemental figures. These results are important because they identify a new dimension to LT scope of action, which might explain some of the sequelae to ETEC exposure, including enteropathy and nutrient malabsorption. Below I list a few points that I hope the authors will consider toward improving the manuscript.

Figure 1, The structure showing the mutation is of limited value to the reader here unless other functionally relevant motifs in LT are identified and labeled.

We appreciate this suggestion, and we have amended the figure to identify the functionally relevant domains of the LT molecule.

Also, perhaps Figure 1 could be combined with figure 2 as thematically they are highly related. Figure 2, The color coding in the cartoon and in the expression data is a confusing, as yellow is used to denote both microvillar adhesion factors as well as higher expression?

We have modified figure 2 by altering the color coding in the figure to make it distinct from the actual expression data in the heat map. We have retained figure 1 as a separate figure to introduce the concept that *many* genes (not just those involving brush border biogenesis identified in figure 2) are altered by LT.

Figure 3, Control villin staining in B panel does not look like one would expect; the yellow signal appears throughout the entire cell rather than the usual enrichment at the apical surface. This point also calls into question the complete lack of villin signal shown in the LT treated cells. Moreover, a complete lack of villin signal does not line up with the fold-change in villin intensity data shown in the plots of the right. The authors need to revisit this analysis, and perhaps stain for additional brush border structural proteins identified in their RNAseq data, to provide more points of validation.

We completely agree with the reviewer. Clearly, villin expression in the LT-treated samples is not zero, as might have been suggested by what came through in the original images obtained at 72 hours rather than after overnight treatment. In the revised figure we show the signals obtained after overnight treatment with LT and have normalized the villin signal at the apical membrane to the cytosolic signal. We have also included villin immunoblots (supplemental figure 2) to further illustrate the impact of LT on villin production.

Figure 3, In this and other figures microvillus length is shown with units as $\text{nm} \times 10^3$? The exponential notation here is needless and confusing for the reader. Just change the units to μm . At the reviewer's suggestion we have removed the exponential notation but expressed the findings in nm to keep the same units of measurement found in the original TEM images.

Figure 5, what is the difference between the normalization shown in panel C versus panel D? These seem redundant. As these data were somewhat redundant, we have simplified the figure by simply retaining the panel (now figure 5c) showing the Villin intensity normalized/enterocyte.

In more than one figure, the intensity of the villin signal is expressed relative to the DAPI nuclear stain. Unfortunately, in sectioned intestinal tissue, the nucleus might not be captured/well represented in the image, which in turn would impact the denominator in this normalization scheme. Because villin signal is normally apically enriched, another well accepted approach would involve normalizing the apical signal relative to the cytoplasmic or even whole cell levels. In this case, the ETEC treated ratio would be closer to 1 whereas the control would be much higher than 1.

The point raised by the reviewer regarding the DAPI nuclear stain is certainly valid. For this reason, we also normalized the data per individual enterocytes the borders of which were easily defined microscopically. In the end both ways of analyzing the data bring us to the same conclusion, that villin expression *in vivo* is also compromised.

Figure legends ask the reader to view the individual panels in a series of images e.g. “clockwise from left” - this is confusing and simply not clear. When showing a series of images, please label each image with a letter and refer to that letter in the figure legend.

We have amended the figure legends accordingly.

Reviewer #2 (Remarks to the Author):

The focus of this manuscript is the impact of enterotoxigenic *Escherichia coli* (ETEC) heat-Labile Toxin (LT) on intestinal epithelial cells. The authors demonstrate that LT has broad impacts on the transcriptomes of Caco-2 cells and human ileal enteroids. Consistent with its ability to decrease expression of genes involved in microvilli biogenesis, LT caused shortening and distortion of microvilli in enteroids, and in neonatal mouse intestines. Maternal vaccination with LT blocked intestinal brush border disruption in ETEC- challenged neonatal mice.

Overall, the approach and methodology are sound, and the observations are noteworthy and novel. The findings add to our knowledge of LT-induced alterations of the intestinal mucosa. The following are some recommendations to improve the manuscript and bolster the findings.

1. The manuscript primarily focuses on alterations in RNA abundance (RNAseq, qRT-PCR), with correlative protein data only for villin (confocal images) and HNF4 γ (Western blots, confocal). The findings will be considerably strengthened by western blot data for additional proteins (brush-border components, SLC19A3, SP1, SMAD4).

In the revised manuscript we have included immunoblotting data for villin (supplemental figure 2), SLC19A3 (supplemental figure 3a), and SP1 (supplemental figure 3b) We have also provided additional transcriptional reporter data for SMAD4. (supplemental figure_6B).

2. What was the rationale for the chosen LT concentration/treatment duration (100 ng/ml for 18 hrs)? LT (50ng/ml for 16 hrs) was previously reported to induce >40% apoptosis of Caco-2 cells (Lu et al, 2017; <https://doi.org/10.3389/fcimb.2017.00244>). Was there significant host cell death with the treatment condition used in the current study at 18 hours (or 72 hours in Fig. 3)?

The reviewer raises an important issue. We did not observe significant cell death in the monolayers. The dose used in our experiments was determined empirically in earlier studies of the induction CEACAM expression by LT (Sheikh, et al PNAS 2020¹: <https://www.ncbi.nlm.nih.gov/pubmed/33139570>), which served as an important benchmark for alterations in gene expression. This dose reliably induces increases in cAMP in both Caco-2 cells and small intestinal enteroids. While we have not specifically examined apoptosis as in the paper by Lu, et al², we should note that we were careful to use only preparations of LT that were known to be free of LPS, a known inducer of apoptosis and other pathways as this could serve to confound observation of any alterations attributed to LT. LPS-free preparations of LT and mutant E211K LT used in the RNAseq experiments were kindly provided by Dr. John Clements of Tulane University (to whom we are also most grateful for pointing out that many laboratory and commercial preparations of LT typically contain *significant* amounts of endotoxin unless specific steps are taking to ensure its removal. We have acknowledged Dr. Clements as the source of LT in the revised manuscript.)

3. Lines 488 & 504: Please indicate if both male and female pups/neonates were included in the study.

Yes, both male and female neonates were used throughout. We have amended the manuscript accordingly.

4. Figure 1 legend: “heat-labile toxin relative to untreated cells or cells treated with the biologically inactive mLT”. The use of “or” in this sentence is confusing: is this intended to be 3-way comparison (LT vs. mLT vs. untreated)? Also, the X-axis label “untreated/mLT” seems to suggest a ratio. Is this the intention? Should the first line of the X axis labels be identical for 1A and 1B? Also, please specify the comparator (differential gene expression/higher/lower relative to ...?) in Supplemental Table 3.

We thank the reviewer for pointing this out. In these initial experiments with Caco-2 cells, we treated replicates (n=2) with LT, mLT (n=2) and compared these to untreated cells (n=2). Because the expression profiles of cells treated with the mutant, inactive toxin were identical to the untreated cells, they were combined (n=4 x-axis) in comparisons against LT treated cells (n=2, y-axis). We have amended the x axis of figure 1a, and the legend to clarify this.

We have likewise amended the legend of supplemental table 3.

5. Figure 2B (also Fig. 4, Fig 7A): For replicates, it appears that n = 2 for each treatment for Caco-2 cells (sham, mLT and LT), and n = 3 for ileal enteroids for the RNAseq data. Is this correct? Please specify in figure legend as appropriate. Also, were enteroids also treated with mLT? If so, include this data in the figure for consistency.

Yes this is correct. Because we did not observe any significant differences in the transcriptomes of mLT and untreated (ø) cells, we omitted mLT from the treatment of enteroids permitting additional replicates (n=3) comparing LT to untreated cells.

It would be instructive to know if the decreases in ESPN and EBP50 are also seen with mLT treatment (as in mouse tissues in Fig. 5; please see next comment).

Actually figure 5 does not involve mLT, but a *toxin deletion mutant* of the bacteria (strain jf4763) in which genes for both LT and ST have been deleted. Therefore, the changes observed here are due to exposure to the bacteria in the absence of any toxin. While there does appear to be some suppression of ESPN and EBP50 with the toxin deletion mutant these changes alone were insufficient to perturb the microvilli (figure 5d,e).

6. Figure 3: Line 114-115 indicates that the inactive toxin failed to induce significant changes in the transcriptome of Caco-2 cells, but it appears that the inactive toxin was almost as effective as the active molecule (please provide statistics for mLT vs. PBS control) in decreasing the expression of ESPN and EBP50. This should be addressed. (see above)

7. Figure 5: (5A) What was the rationale for using a lacZYA::Kmr derivative instead of the parent H10407? We have used jf876 in multiple colonization experiments. This strain contains a kanamycin cassette in the lacZYA permitting us to examine colonization by plating bacteria onto LB plates kanamycin and Xgal. This strain colonizes as well as the wild type H10407. We did not see obvious differences in colonization of suckling mice in these studies relative to the non-toxicogenic derivative which is also kanamycin-resistant.

(5G) jf570 is not listed in Supplemental table 1 – should it be jf571? jf571 is a duplicate stock of jf570. We have amended table 1 to eliminate confusion here.

It is stated in Supplemental Table 1 that jf4763 was generated as part of the current study. If so, please provide details about strain construction in the Methods section.

We appreciate this observation by the reviewer, and have added details to the supplemental table and the methods.

8. Figure 6: Please include data on bacterial burden in stool/tissues for all animals. It would be informative to know if the pups were protected against microvillus disruption despite comparable bacterial burdens (or if, somehow, maternal LT vaccination also curtailed ETEC colonization/persistence).

While we certainly agree that this might have been informative, as we have previously demonstrated that LT contributes to intestinal colonization in adult mice^{3,4}. However, we did not collect data on colonization in these studies for several reasons. First the amount of fecal material produced by suckling mice is very scant and would not be sufficient to monitor colonization. In addition, because we were monitoring changes in body weight in challenged mice, we would only have been able to generate end-point data for the burden in tissues at the time the animals were sacrificed.

9. Supplemental Figure 4: This model is speculative and is not adequately supported by the data provided. Some experimental validation is warranted: Are LT-mediated transcriptional changes (or brush border alterations) reversed in transfected cells expressing HNF4 and/or SMAD4? Does PKA inhibition inhibit LT-mediated changes in HNF4/SMAD4 and downstream targets? Is there an increase in phosphorylated HNF4 in LT-treated cells?

The model was intended to be speculative, and we have clarified this in the revision. Our intent in this figure was to (1) demonstrate that SMAD4 transcription as well as SMAD4-mediated transcription was impacted (2) illustrate how our present findings relate to what is already known regarding the relationships of PKA⁵, HNF4⁶, SMAD4⁷ and enterocyte development. We certainly agree that additional studies are warranted. While we have also added additional data examining SMAD4-mediated transcription in what is now supplemental figure 6B, additional experimentation suggested by the reviewer would still be needed to validate the model. However, we felt that these studies would be beyond the scope of the present studies focused on the impact of LT on the brush border.

Minor issues:

1. The arbitrary use of small case letters at the beginning of sub-headings etc. is distracting. Please adhere to standard English conventions throughout the document.

See amended manuscript.

2. Where possible, please include toxin concentration and treatment duration in the figure legends.

See amended manuscript.

3. Line 114-115: For precision, please delete "binding of". Amended

4. Line 192 – end of Results section: This reverts to data on enteroids/Caco-2 cells. For better cohesion, please consider moving this section before the mouse data (Line 163).

5. Line 352: Please indicate LT source.

This was an important oversight. We appreciate the thorough review and as noted above have included this in the revision.

6. Figure 3 legend: Please specify cell type, LT concentration and treatment duration, number of replicates and number of fields assessed. See amended manuscript.

7. Figure 6: In the schematic, distinguish between maternal treatments (LT) and infant mouse parameters (ETEC infection and tissue collection) - possibly with downward arrows for the former.

See amended figure 6 legend.

Reviewer #3 (Remarks to the Author):

This manuscript provides compelling data that heat-labile toxin (LT) from ETEC is capable of reducing expression of several genes involved in nutrient transport. Functional confirmation of reduced expression of the major thiamine transporter is provided by showing reduced transport of the micronutrient thiamine (3H-thiamine). The authors use complementary Caco-2 IEC lines, primary human intestinal enteroid cultures and in vivo infection models. Mechanistically, the paper implicates a cAMP-PKA mediated suppression of HNF4 target genes involved in microvilli development and transporter gene expression. The paper is well written although the data do not achieve the stated goal of functionally connecting LT infection with ETEC induced enteropathic disease.

The quantification graph for Fig 3B should mention villin to make it easier for the reader to identify the relevance of the figure relative to the immunostaining images in panel 3B. this was included in the amended figure 3 with this revision.

Did the authors confirm that LT treatment did not increase tight junction permeability as this would negatively impact upon epithelial monolayer integrity and transporting function? The reviewer raises an interesting question. Although not reported here, TEER does decrease somewhat in both polarized Caco-2 cells and in enteroids following treatment with LT. Based on what is already known regarding the action of the closely homologous cholera toxin this would certainly be anticipated⁸. (While this may also contribute to development of compromised barrier function frequently observed in children with enteropathy⁹, we felt that an extended exploration of barrier function would be beyond the scope of the present manuscript).

In the in vivo mouse challenge model, a 7-day infection with the LT/ST competent wild-type ETEC induced reduced expression of nutrient transporters and altered microvillus architecture, consistent with the in vitro IEC challenge studies. However, this had no effect on the growth kinetics of the mice, suggesting that ETEC did not compromise development. Seven days may be too short a timeframe to identify an effect on weight loss. In order for the authors to support their hypothesis that ETEC infection can predispose to non- diarrheal and enteropathic pathologies, some evidence of a pathologic event arising from microvillus shortening and/or nutrient deprivation needs to be demonstrated. The absence of such evidence undermines the translational relevance of the study which is the stated major emphasis of this paper.

We completely agreed with the reviewer. We have in the revision included additional data that supports the hypothesis that ETEC contributes to enteropathy at least in part by expression of LT. As deduced by the reviewer, seven days was simply too short to observe any difference in the growth kinetics of mice following ETEC infection (supplemental figure 5A). In the revised manuscript, we demonstrate that only *repeated infection* culminates in altered growth kinetics (supplemental figure 5B) recapitulating multiple observations in young children in developing countries where the risk of enteropathy appears to increase multiplicatively with repeated infections. Furthermore, we demonstrate that LT expression by the infecting strain has a demonstrable impact on growth kinetics (supplemental figure 5C). Although many questions remain, we appreciate the input and the direction provided by the reviewer.

References

1. Sheikh, A., Tumala, B., Vickers, T.J., Alvarado, D., Ciorba, M.A., Bhuiyan, T.R., Qadri, F., Singer, B.B. & Fleckenstein, J.M. CEACAMs serve as toxin-stimulated receptors for enterotoxigenic *Escherichia coli*. *Proc Natl Acad Sci U S A* **117**, 29055-29062 (2020) PMC7682567.
2. Lu, X., Li, C., Li, C., Li, P., Fu, E., Xie, Y. & Jin, F. Heat-Labile Enterotoxin-Induced PERK-CHOP Pathway Activation Causes Intestinal Epithelial Cell Apoptosis. *Front Cell Infect Microbiol* **7**, 244 (2017) PMC5463185.
3. Roy, K., Hamilton, D.J. & Fleckenstein, J.M. Cooperative role of antibodies against heat-labile toxin and the EtpA Adhesin in preventing toxin delivery and intestinal colonization by enterotoxigenic *Escherichia coli*. *Clin Vaccine Immunol* **19**, 1603-1608 (2012) 3485888.
4. Allen, K.P., Randolph, M.M. & Fleckenstein, J.M. Importance of heat-labile enterotoxin in colonization of the adult mouse small intestine by human enterotoxigenic *Escherichia coli* strains. *Infect Immun* **74**, 869-875 (2006)
5. Violette, B., Kahn, A. & Raymondjean, M. Protein kinase A-dependent phosphorylation modulates DNA-binding activity of hepatocyte nuclear factor 4. *Molecular and cellular biology* **17**, 4208-4219 (1997) PMC232274.
6. Chen, L., Luo, S., Dupre, A., Vasoya, R.P., Parthasarathy, A., Aita, R., Malhotra, R., Hur, J., Toke, N.H., Chiles, E., Yang, M., Cao, W., Flores, J., Ellison, C.E., Gao, N., Sahota, A., Su, X., Bonder, E.M. & Verzi, M.P. The nuclear receptor HNF4 drives a brush border gene program conserved across murine intestine, kidney, and embryonic yolk sac. *Nature communications* **12**, 2886 (2021) PMC8129143.
7. Chen, L., Toke, N.H., Luo, S., Vasoya, R.P., Fullem, R.L., Parthasarathy, A., Perekatt, A.O. & Verzi, M.P. A reinforcing HNF4-SMAD4 feed-forward module stabilizes enterocyte identity. *Nat Genet* **51**, 777-785 (2019) PMC6650150.
8. Guichard, A., Cruz-Moreno, B., Aguilar, B., van Sorge, N.M., Kuang, J., Kurkciyan, A.A., Wang, Z., Hang, S., Pineton de Chambrun, G.P., McCole, D.F., Watnick, P., Nizet, V. & Bier, E. Cholera toxin disrupts barrier function by inhibiting exocyst-mediated trafficking of host proteins to intestinal cell junctions. *Cell Host Microbe* **14**, 294-305 (2013) PMC3786442.
9. Amadi, B., Zyambo, K., Chandwe, K., Besa, E., Mulenga, C., Mwakamui, S., Siyumbwa, S., Croft, S., Banda, R., Chipunza, M., Chifunda, K., Kazhila, L., VanBuskirk, K. & Kelly, P. Adaptation of the small intestine to microbial enteropathogens in Zambian children with stunting. *Nat Microbiol* **6**, 445-454 (2021) PMC8007472.

REVIEWERS' COMMENTS

Reviewer #1 (Remarks to the Author):

The authors did a nice job addressing my minor comments. I have no further suggestions.

Reviewer #2 (Remarks to the Author):

The studies in this manuscript demonstrated that enterotoxigenic *Escherichia coli* heat-labile toxin has broad impacts on the transcriptomes of Caco-2 cells and human ileal enteroids. LT decreased expression of genes involved in microvilli biosynthesis and caused shortening and distortion of microvilli enteroids and in neonatal mouse intestines. Maternal vaccination with LT curtailed brush border disruption in ETEC-challenged neonatal offspring. The overall approach and methodology are sound, and the observations are novel and noteworthy. The authors have satisfactorily addressed all the major concerns raised previously.

Reviewer #3 (Remarks to the Author):

While the response of the authors to the concern regarding a possible contribution of tight junction remodeling - and increased permeability - contributing to reduced capacity for nutrient uptake was acceptable, it is recommended that the authors acknowledge this with a sentence in the Discussion so as to provide the appropriate context for readers of additional relevant factors that could contribute to the observed phenotype in vivo.

point-by-point response to the reviewers' comments

We greatly appreciate the thorough review and the input from each of the reviewers in helping us to draft an improved manuscript. Only reviewer three asked for additional changes which we have included in the revised discussion.

Reviewers' comments

Reviewer #1 (Remarks to the Author):

The authors did a nice job addressing my minor comments. I have no further suggestions.

Reviewer #2 (Remarks to the Author):

The studies in this manuscript demonstrated that enterotoxigenic *Escherichia coli* heat-labile toxin has broad impacts on the transcriptomes of Caco-2 cells and human ileal enteroids. LT decreased expression of genes involved in microvilli biosynthesis and caused shortening and distortion of microvilli enteroids and in neonatal mouse intestines. Maternal vaccination with LT curtailed brush border disruption in ETEC-challenged neonatal offspring. The overall approach and methodology are sound, and the observations are novel and noteworthy. The authors have satisfactorily addressed all the major concerns raised previously.

Reviewer #3 (Remarks to the Author):

While the response of the authors to the concern regarding a possible contribution of tight junction remodeling - and increased permeability - contributing to reduced capacity for nutrient uptake was acceptable, it is recommended that the authors acknowledge this with a sentence in the Discussion so as to provide the appropriate context for readers of additional relevant factors that could contribute to the observed phenotype in vivo.

We have added two additional references and alluded to the potential impact of LT on barrier function in the revised discussion on lines 326-329.